# APEX2-Mediated Proximity Labeling Resolves the DDIT4-Interacting Proteome

**DOI:** 10.3390/ijms23095189

**Published:** 2022-05-06

**Authors:** Marianna Naki, Olga Gourdomichali, Katerina Zonke, Fedon-Giasin Kattan, Manousos Makridakis, Georgia Kontostathi, Antonia Vlahou, Epaminondas Doxakis

**Affiliations:** 1Center of Basic Research, Biomedical Research Foundation, Academy of Athens, 11527 Athens, Greece; mnaki@bioacademy.gr (M.N.); ogourdom@bioacademy.gr (O.G.); katerinazonke@gmail.com (K.Z.); fkattan@bioacademy.gr (F.-G.K.); mmakrid@bioacademy.gr (M.M.); gkontostathi@bioacademy.gr (G.K.); vlahoua@bioacademy.gr (A.V.); 2Department of Physiology, National and Kapodistrian University of Athens (NKUA), 11527 Athens, Greece; 3Department of Biology, National and Kapodistrian University of Athens (NKUA), 15784 Athens, Greece; 4Department of Biological Applications and Technology, Faculty of Health Sciences, University of Ioannina, 45110 Ioannina, Greece

**Keywords:** DDIT4, APEX2, proximity labeling, interactome, proteomics, LC-MS/MS, acute stress

## Abstract

DNA damage-inducible transcript 4 (DDIT4) is a ubiquitous protein whose expression is transiently increased in response to various stressors. Chronic expression has been linked to various pathologies, including neurodegeneration, inflammation, and cancer. DDIT4 is best recognized for repressing mTORC1, an essential protein complex activated by nutrients and hormones. Accordingly, DDIT4 regulates metabolism, oxidative stress, hypoxic survival, and apoptosis. Despite these well-defined biological functions, little is known about its interacting partners and their unique molecular functions. Here, fusing an enhanced ascorbate peroxidase 2 (APEX2) biotin-labeling enzyme to DDIT4 combined with mass spectrometry, the proteins in the immediate vicinity of DDIT4 in either unstressed or acute stress conditions were identified in situ. The context-dependent interacting proteomes were quantitatively but not functionally distinct. DDIT4 had twice the number of interaction partners during acute stress compared to unstressed conditions, and while the two protein lists had minimal overlap in terms of identity, the proteins’ molecular function and classification were essentially identical. Moonlighting keratins and ribosomal proteins dominated the proteomes in both unstressed and stressed conditions, with many of their members having established non-canonical and indispensable roles during stress. Multiple keratins regulate mTORC1 signaling via the recruitment of 14-3-3 proteins, whereas ribosomal proteins control translation, cell cycle progression, DNA repair, and death by sequestering critical proteins. In summary, two potentially distinct mechanisms of DDIT4 molecular function have been identified, paving the way for additional research to confirm and consolidate these findings.

## 1. Introduction

DNA damage-inducible transcript 4 (DDIT4, also known as REDD1 and RTP801) is a highly conserved stress-response protein of 232 amino acids [1]. It is developmentally regulated and ubiquitously expressed at low levels in most adult tissues [1,2]. Multiple stress signals induce the expression of DDIT4 that are classified into three broad categories: stress-related hormones such as glucocorticoids and aldosterone [3,4]; energy stressors such as hypoxia, fasting, and glucose deprivation [1,3,5]; and cellular stressors such as DNA damage, osmotic and endoplasmic reticulum (ER) stress, and excessive reactive oxygen species (ROS) [2,5,6,7,8]. DDIT4 is mainly localized in the cytoplasm, but it can also be detected in the plasma membrane, mitochondrial-associated ER membranes, and the nucleus [2,9,10,11].

The best-known cellular function of DDIT4 is the inhibition of the mechanistic target of rapamycin complex 1 (mTORC1) [12,13], which regulates cell growth by promoting ribosomal and mitochondrial biogenesis, as well as protein synthesis (reviewed in [14]). The activation of mTORC1 is by mitogens (insulin, growth factors), nutrients (amino acids), and energy levels via the phosphoinositide 3-kinase/ protein kinase B (PI3K/AKT) signaling pathway [14]. Mechanistically, AKT phosphorylates the upstream repressor of mTORC1, tuberous sclerosis 1/2 complex (TSC1/2), which is then sequestered by 14-3-3 proteins [15,16]. This consequentially reduces the available GTPase-activating protein (GAP) activity of TSC2 towards the Ras homolog enriched in the brain (RHΕΒ) that remains in its active GTP-bound form, thereby speeding up catalysis reactions by mTORC1 by allosterically realigning active-site residues on its kinase domain [17]. In response to stress, DDIT4 is thought to activate TSC1/2 [12] by two distinct mechanisms. In the first, DDIT4 sequesters 14-3-3 proteins away from TSC2, allowing TSC2 to repress mTORC1 [9]. However, structure-based docking, conservation, and functional analyses suggested that such interaction is improbable [18]. Alternatively, DDIT4 interacts with protein phosphatase 2A (PP2A), facilitating its activity towards the Thr308 residue of AKT, partially inactivating it [19].

Current thinking postulates that DDIT4 serves as a transient austerity messenger critical for the physiological adaptation to stresses, provided that its expression is limited to a short, timely window. In contrast, sustained expression of DDIT4 drives metabolic dysregulation and apoptosis (reviewed in [20]). Expectedly, DDIT4 is an early biomarker for many human pathologies, including neurodegenerative diseases, depression, diabetes, cancer, and inflammatory diseases [20].

The partial crystal structure of DDIT4 has revealed a sandwich structure with two antiparallel α-helices followed by a mixed β-sheet containing four β-strands [18]. A conserved surface patch is required for DDIT4 signaling. It is formed by two distinct regions, a loop between helix 2 and strand 1, and six residues of strand 4 [18]. Currently, we know little about DDIT4′s interacting partners in either naive or stress conditions that limit our understanding of its molecular function. Proximity labeling is a technique in which a labeling enzyme fused to the protein of interest marks the protein’s interaction network in vivo, enabling further ex vivo analysis. The main advantage of this technique is that it is performed in living cells where all compartments are intact. Further, this method can also detect transient and dynamic interactions in contrast to biochemical methods such as co-immunoprecipitation (co-IP) and protein pull-down, which rely on only direct and stable interactions. To conduct proximity labeling, a second-generation highly active ascorbate peroxidase (APEX2) was linked to DDIT4 [21]. APEX2 uses hydrogen peroxide (H_2_O_2_) as an oxidant to catalyze the one-electron oxidation of a cell-permeable, biotin-tyramide substrate to generate highly reactive and short-lived biotin phenoxyl radicals that label aromatic amino acids in proteins within 20 nm of the enzyme [21]. Using this technique, all proteins proximal to DDIT4 were labeled in unstressed and acute stress conditions. Subsequently, they were identified using mass spectrometry (MS) and validated by combining co-IP and immunoblotting studies. The results highlight the interaction of DDIT4 with moonlighting cytoskeletal and ribosomal proteins, shedding light on the principal molecular mechanisms underlying its function.

## 2. Results

### 2.1. Construction of a Functional DDIT4–APEX2 Fusion Protein for Proximity Labeling

To produce the fusion protein for proximity biotin labeling, the DNA fragment of APEX2 was linked via a flexible (GGGS)3 linker to the C-terminus of DDIT4 DNA and inserted into the pAAV expression vector that utilizes a CAGGS promoter and a WPRE sequence for efficient transcription and translation, respectively. To test whether the fusion protein is functional, SK-N-SH cells were transfected with APEX2 or DDIT4–APEX2 plasmids, and at 36 h, cells were incubated with biotin phenol with or without sodium arsenite for 45 min, before activating APEX2 by a short pulse of H_2_O_2_ to induce labeling (Figure 1A,B). The cells were then immediately lysed and the biotinylated proteins were isolated with streptavidin magnetic beads. Western blots probed with NeutrAvidin-HRP confirmed APEX2 labeling driven by APEX2 and DDIT4–APEX2 plasmids; however, labeled proteins were significantly less in the DDIT4–APEX2 conditions. To minimize the possibility that this is due to the linker sequence, another construct was prepared, in which a much longer rigid linker (GGAEAAAKEAAAKAAPAEAAAKEAAAKA) was inserted between DDIT4 and APEX2. The assay was repeated and revealed that the plasmid with the rigid linker yielded more labeled proteins, presumably because it effectively separated the two proteins, minimizing the interference in biological activity caused by their interaction; thus, it was chosen for subsequent experiments (Figure 1C). To determine whether the fusion protein preserves the subcellular distribution of DDIT4, immunocytochemistry with antibodies against DDIT4 was performed on SK-N-SH cells transfected with DDIT4 or DDIT4–APEX2 constructs in unstressed and acute stress conditions. Figure 1D shows that, while APEX2 labeling showed diffuse subcellular distribution regardless of whether cells were treated with sodium arsenite or not, DDIT4 and DDIT4–APEX2 distribution was strongly cytosolic.

### 2.2. Proteomic Identification of DDIT4 Partners in Unstressed and Acutely Stressed Conditions

Proteomics analysis (LC-MS/MS) was used to identify all DDIT4 immediate neighboring proteins in unstressed and acutely stressed (sodium arsenite, 45 min) neuroblastoma cells. Sodium arsenite was used as a well-documented stressor mobilizing RNA binding proteins to stress granules and a known inducer of DDIT4 expression [10]. For each condition (unstressed or acutely stressed), four independent LC-MS/MS analyses were performed. Figure 1E displays the overlapping hits for the two DDIT4–APEX2 labeling conditions. In total, 316 proteins were detected across the DDIT4–APEX2 experiments by MS in the unstressed condition, and after filtering out the dataset for non-specific labeling by APEX2 alone (571 proteins), 44 proteins were unique to DDIT4 (Appendix A). In the acutely stressed condition, DDIT4–APEX2 labeled a similar number of proteins (337 proteins), and after filtering out the APEX2 dataset (446 proteins), 81 proteins were unique to DDIT4 (Appendix A).

### 2.3. Bioinformatic Analysis of DDIT4 Partners in Unstressed and Acutely Stressed Conditions

To analyze the protein lists obtained from the proteomics studies, we initially used the WebGestalt analysis toolkit. In the unstressed condition, DDIT4 partners were almost equally distributed between the cytosol, membranes, nucleus, and protein complexes (Figure 2A). Using enrichment analysis, it was revealed that the partners are primarily associated with the ‘intermediate filament cytoskeleton’ [Enrichment/Ratio (E/R) 13, FDR 5.7 × 10^−9^] and the ‘cytosolic large ribosomal subunit’ (E/R 24.7, FDR 5.3 × 10^−2^) (Figure 2B). In the acutely stressed condition, DDIT4 partners were predominantly cytosolic (Figure 2C). Similar to the unstressed condition, enrichment analysis of their localization during stress revealed the ‘intermediate filament cytoskeleton’ (E/R 18.7, FDR 9.8 × 10^−13^) and the ‘cytosolic ribosome’ (E/R 18.1, FDR 1.3 × 10^−5^) as the most overrepresented categories (Figure 2D).

Next, the ‘Molecular Function’ of the pulled-down proteins was analyzed. DDIT4 partners displayed mostly ‘protein binding activity’ in unstressed conditions and were enriched for ‘structural molecular activity’ (E/R 5.85, FDR 8.5 × 10^−3^) (Figure 3A,B). In the sodium-arsenite-treated condition, DDIT4 partners also displayed mainly ‘protein binding’ function and were enriched for ‘structural molecular activity’ (E/R 5.85, FDR 8.5 × 10^−3^), ‘structural constituents of ribosomes’ (E/R 9.7, FDR 7.6 × 10^−6^), and ‘RNA binding activity’ (E/R 3.5, FDR 5.4 × 10^−9^) (Figure 3C,D). Based on these data, DDIT4-interacting partners’ overall ‘molecular function’ is similar in the control and acute stress conditions, even though only seven proteins were shared between the two protein lists (Appendix A).

To visualize the biological terms for both unstressed and sodium-arsenite-treated conditions in a functionally grouped network, the two protein lists were combined and the ClueGo plugin was run on Cytoscape. Figure 4 shows the interconnection of the different GO ‘biological processes’ and those not directly related to others. Further, Table 1 lists the IDs of the DDIT4-interacting proteins associated with each GO ‘biological process’. Most enriched categories included ‘intermediate filament organization’ (FDR 1.5 × 10^−6^), ‘cytoplasmic translation’ (FDR 2.7 × 10^−6^), ‘glycolytic process’ (FDR 1.2 × 10^−2^), and ‘ribosomal large subunit biogenesis’ (FDR 2.3 × 10^−2^).

### 2.4. Identification of DDIT4 Directly-Interacting Proteins

Co-immunoprecipitation analysis for representative proteins was conducted to corroborate proteomics findings and identify DDIT4 directly-interacting proteins. DDIT4 displayed a strong affinity for structural maintenance of chromosomes 3 (SMC3), ectodermal-neural cortex 1 (ENC1), vimentin (VIM), E2F7, intracellular Notch receptor 2 (NOTCH2-IC), keratin 17 (KRT17), and prohibitin 1 (PHB1) (Figure 5). Two other proteins, tyrosine 3-monooxygenase/tryptophan 5-monooxygenase activation protein theta (YWHAQ, 14-3-3θ), and NME/NM23 nucleoside diphosphate kinase 1 (NME1), were only faintly co-immunoprecipitated with DDIT4, indicating that their interaction is indirect or weak (not shown).

## 3. Discussion

DDIT4 is a key stress-regulated protein implicated in various cellular processes due to its interaction with proteins that impact protein homeostasis and metabolic function. There has been no study that captures DDIT4′s interacting partners. Here, recognizing the importance of both subcellular complexity and spatial context for the biological function of proteins, the APEX2 labeling method was used to identify all of its proximity partners in living cells. DDIT4 partners were identified in unstressed and acute stress conditions, and a selection of them were probed for direct interaction with DDIT4. It should be emphasized that the DDIT4–APEX2 hybrid labeled any protein within a radius of approximately 20 nm, not just those directly complexed with DDIT4. This enabled the resolution of the molecular neighborhood that is potentially more physiologically relevant than direct only interactions. Further, it should be pointed out that this study was conducted on a single cell population, ex vivo, that may only partially reflect the total proteome available to DDIT4 for interactions.

In terms of the data itself, there was a quantitative but not functional distinction between the proteomes obtained in unstressed and acutely stressed conditions. DDIT4 had twice the number of interaction partners during acute stress, presumably reflecting the state in which DDIT4 takes on a central role in responding to the stress insult. Additionally, while there was little overlap between the two protein lists, as thousands of proteins are induced or repressed during acute stress, the molecular functions of DDIT4′s protein partners were very similar in the two conditions, indicating that the DDIT4 function is directed toward specific classes of proteins to fine-tune repair responses [22]. Two categories dominated the protein lists in unstressed and stressed conditions; the intermediate filaments (IFs, also known as keratins, KRT) with over 21 members and the ribosomal proteins (RPs) with 13 members.

Keratins function through interaction with structural proteins and proteins involved in cell signaling, particularly those associated with stress responses, apoptosis, and cell proliferation [23]. For instance, for keratin 17 (KRT17), one protein pulled-down and co-IPed with DDIT4 is required for the serum-dependent relocalization of 14-3-3 from the nucleus to the cytoplasm, and the concomitant stimulation of mTOR activity and cell growth [24]. Further, KRT17 is induced following DNA damage and promotes cell survival [25]. Similarly, phosphorylation of KRT18, another protein that was pulled-down with DDIT4, promotes its binding to 14-3-3zeta and stimulates mitosis through the activation of 14-3-3 signaling in the cytosol [26]. Moreover, the interaction of KRT18 with TNFR1-associated death domain protein (TRADD), an indispensable adaptor for TNFR signaling, rendered cells more resistant to TNF-induced apoptosis [27,28]. Analogous findings on signaling have been reported with several other KRTs identified in this study (see KRT8 and VIM [29,30,31,32]). Hence, it is conceivable that one role of DDIT4 is to control mTORC1 signaling via binding to KRTs and mediate their direct interaction with 14-3-3 proteins. Further, additional, more direct routes to control mTORC1 activation may be available, as DDIT4–APEX2 labeling also detected protein phosphatase 2 scaffold subunit A (PPP2R1A) in stress-induced cells.

Ribosomal proteins are essential for ribosome biogenesis and protein synthesis, the cornerstone mechanism for cell growth and proliferation regulated by mTORC1. However, they are also moonlight proteins (a single protein with multiple functions) with extra-ribosomal functions regulating diverse cellular processes, including the cell cycle, DNA repair, genome integrity, cellular proliferation, stress, apoptosis, and autophagy [33]. Hence, they are intensively investigated in cancer therapy and diagnosis. Many RPs identified here as DDIT4-interacting partners have established such functions. RPS13 promotes the growth and cell cycle progression of cancer cells by inhibiting the expression of tumor suppressor p27 [34]. RPL23 sequesters nucleophosmin (NPM) that is required for tumor suppressor myc-associated zinc finger protein 1 (Miz1) activation [35]. RPL10 aids cancer progression by regulating cellular ROS levels in mitochondria [36]. In response to different cellular stresses, RPS7, RPS25, RPS27, and RPL26 activate tumor suppressor p53 by forming a complex with the E3 ubiquitin ligase murine double minute 2 (MDM2) that facilitates typically proteasomal degradation of p53 [37,38,39]. Similarly, upon excessive arsenite treatment, RPS7 recruits MDM2 to prevent degradation of stress-induced DNA-damage inducible (GADD) 45a protein [40]; multifunctional GADD45a subsequently induces cell cycle arrest, contributes to DNA repair, and activates the JNK pathway [41]. Further, upon UV-irradiation or genotoxic stress, RPS3 induces apoptosis by activating the caspase-3/8 cascade and JNK pathway [42]. Hence, DDIT4′s other main role is to participate in RP-mediated stress responses in concert with mTORC1 signaling regulation.

In conclusion, this study provides a system-level view of the DDIT4 interactome and the possible means of regulating stress signaling. Mechanistic studies will next confirm the impact of DDIT4′s interactions in different cellular contexts.

## 4. Materials and Methods

### 4.1. Antibodies

The rabbit polyclonal anti-DDIT4 (#10638-1-AP), anti-ENC1 (#15007-1-AP), anti-SMC3 (#14185-1-AP), anti-KRT17 (#17516-1-AP), anti-PHB1 (#10787-1-AP), anti-E2F7 (#24489-1-AP), anti-NOTCH2 (#28580-1-AP), anti-YWHAQ/14-3-3Θ (#14503-1-AP), and anti-NME1 (11086-2-AP) were from Proteintech (Chicago, IL, USA); the mouse monoclonal anti-GAPDH (sc-365062) antibody and normal rabbit IgG (sc-2027) were obtained from Santa Cruz Biotechnology (Santa Cruz, Dallas, TX, USA); the goat anti-rabbit FITC-conjugated secondary antibody (Alexa Fluor 488, #A27034) and the streptavidin-Alexa 568 conjugate (#S11226) were from ThermoFisher (Waltham, MA, USA); the mouse (#7076) and rabbit (#7074) HRP-conjugated secondary antibodies were from Cell Signaling Technologies (Danvers, MA, USA). The anti-VIM (ab92547) was obtained from Abcam (Cambridge, UK).

### 4.2. Generation of DNA Constructs

The murine DDIT4 CDS+3′UTR, human DDIT4 CDS, and APEX2 CDS (without FLAG and NES tags) were amplified by PCR using the proofreading Phusion polymerase (ThermoFisher) from Neuro-2a cells, SK-N-SH cells, and pcDNA3 FLAG-APEX2-NES plasmid (Addgene #49386), respectively (see Appendix A for primer sequences). The PCR products were then cloned between the HindIII and BamHI restriction sites on the paavCAG-pre-mGRASP plasmid (Addgene #34911) using the HiFi system (NEB, Massachusetts, CA, USA). Two alternative DDIT4 fusion proteins were prepared that contained the full-length sequences of DDIT4 and APEX2 and either a short flexible (GGGS)3 or rigid (GGAEAAAKEAAAKAAPAEAAAKEAAAKA) linker sequence in between. Sanger sequencing verified the DNA sequence of all constructs at CeMIA SA (Larisa, Greece).

### 4.3. In Situ Labeling of DDIT4 Interactors Mediated by APEX2-Mediated Biotinylation

Neuroblastoma SK-N-SH cells were cultured in high-glucose DMEM (#D6429, Sigma-Aldrich, St. Louis, MO, USA) supplemented with 10% fetal bovine serum (FBS) (#16000044, ThermoFisher) and 1% penicillin/streptomycin (#P4333, Sigma-Aldrich). Cells were maintained at 37 °C in a humidified 5% CO_2_ incubator (Thermo Forma, ThermoFisher). SK-N-SH cells were transfected at plating with APEX2 or DDIT4–APEX2 expression plasmids using the JetOptimus transfection reagent (Polyplus, Illkirch, France). To maintain labeling stoichiometry and a similar low-level APEX2-associated toxicity, the plasmid encoding APEX2 alone was transfected at a 35% lower concentration. Thirty-six hours after transfection, cells were supplemented with 1 mM biotin tyramide (#LS-3500, Iris Biotech, Marktredwitz, Germany) in the presence, or not, of 330 μM NaAsO_2_ (#S7400, Sigma-Aldrich) (reduced to 100 μΜ for immunofluorescence staining experiments due to the cells’ poor attachment) for 45 min at 37 °C. Afterward, H_2_O_2_ was added at a final concentration of 1 mM for exactly 1.5 min at room temperature (RT). The reaction was quenched by adding Trolox (sc-200810) and sodium ascorbate (sc-215877) (both from Santa Cruz Biotechnology) dissolved in PBS to a final concentration of 5 mM and 10 mM, respectively. Cells were washed twice more with quenching solution and either fixed for 15 min with 4% (*w*/*v*) paraformaldehyde (#A11313, Alfa Aesar, Haverhill, MA, USA) for immunofluorescent staining experiments or lysed with RIPA solution (see below) for the affinity capture assay.

### 4.4. Immunofluorescence Staining

Paraformaldehyde fixed SK-N-SH cells cultured on poly-D-lysine-treated coverslips were permeabilized with PBS containing 0.5% Triton X-100 for 10 min at RT. Cells were subsequently incubated for 1 h with blocking solution consisting of 3% BSA, 0.02% Triton X-100 in PBS (PBST) at RT and then probed overnight at 4 °C with a primary antibody against DDIT4 diluted in blocking solution at 1:75 in a hybridization chamber. The next day, cells were washed thrice with PBS and incubated with either goat anti-rabbit FITC-conjugated secondary antibody (diluted at 1:500) or streptavidin-Alexa 568 conjugate (diluted at 1:2000) in blocking solution (1% BSA in PBS) for 1 h at RT. Cells were washed twice with PBS, after which DAPI (#D9542, Sigma-Aldrich) was added for 3 min, followed by another two rounds of PBS wash. Coverslips were mounted on slides using Vectashield (#H-1700, Vector labs, Burlingame, CA, USA). Confocal imaging was performed using a Leica inverted confocal laser scanning microscope. Images were acquired using Leica LAS AF software through a 60× oil immersion objective. All fluorescence images between different sample groups were obtained using identical settings.

### 4.5. Preparation of Whole Protein Extracts and Affinity Capture of Biotinylated Proteins

Following the APEX2 labeling reaction, SK-N-SH cells were washed twice with a quenching solution and resuspended in 200 μL of ice-cold RIPA lysis buffer containing 25 mM Tris pH 7.5, 150 mM NaCl, 1% Triton X-100, 0.16% sodium deoxycholate, 0.16% SDS, and supplemented with 1.5 mM EDTA, 5 mM Trolox, 10 mM L-ascorbate and protease inhibitors (cOmplete, Roche, Basel, Switzerland). After incubation for 30 min in a rotor at 4 °C, cell suspensions were centrifuged for 15 min at 14,000× *g* at 4 °C, and the supernatants (whole-cell lysates) were transferred into new tubes. Total protein amounts were then quantified by the Bradford Assay (BioRad Laboratories, Richmond, CA, USA).

Streptavidin magnetic particles (#11641786001, Roche) were equilibrated with RIPA lysis buffer by two washes. Each lysate was incubated with 90 μL of bead slurry in microcentrifuge tubes with rotation for 2 h at RT. The beads were subsequently washed twice with 1 mL RIPA lysis buffer, once with 1 mL of 1 M KCl, once with 1 mL of 100 mM sodium carbonate, twice with 1 mL of 2 M urea, and twice with 1 mL of RIPA lysis buffer. Biotinylated proteins were then eluted by incubating the bead slurry with 100 μL of 2 M thiourea, 6 M urea, 1% SDS, and 3 mM biotin in ddH_2_O for 15 min at RT followed by another 15 min incubation at 98 °C.

### 4.6. Western Blot Analysis

Immunoblotting was carried out as previously described [43]. Briefly, equal amounts of whole-cell extracts or equal volumes of pull-down material were separated by 12% SDS-PAGE under denaturing conditions and transferred to a nitrocellulose membrane (Protran; Amersham/Merck, St. Louis, MO, USA). Nitrocellulose membranes were then blocked with Tris-buffered saline (TBS) containing 5% non-fat milk and 0.1% Tween-20 for 1 h at RT and afterward were probed with the respective primary antibodies. All primary antibodies were diluted in blocking buffer at a final concentration of 1:1000, except for the NeutrAvidin–HRP conjugate prepared at a final concentration of 1:2000. All secondary HRP-conjugated antibodies were used in a 1:2000 dilution. The immunoreactive bands were visualized with the enhanced chemiluminescence (ECL) method using the Clarity substrate (BioRad, Hercules, CA, USA).

### 4.7. Liquid Chromatography-Tandem Mass Spectrometry

Twelve biologically independent labeling experiments were conducted for each condition (unstressed or acutely stressed), three replicates were merged, and four independent LC-MS/MS analyses were performed.

#### 4.7.1. Sample Preparation

As mentioned earlier, biotinylated proteins were eluted after APEX labeling reactions with a buffer composed of 2 M thiourea, 6 M urea, 1% SDS, and 3 mM biotin. Samples were subjected to buffer exchange (with 50 mM NH_4_HCO_3_) and, at the same time, concentrated to a final volume of 20 μL by utilizing Amicon centrifugation filters with 3 kDa molecular weight cutoff (MWCO). All of the available volume of each sample (20 μL) was analyzed in SDS-PAGE (5% stacking, 12% separating) with the GeLC-MS method, as previously described [44]. Briefly, electrophoresis was terminated when samples entered the separating gel. Gels were fixed in 30% methanol and 10% acetic acid for 30 min, washed thrice with water (3 × 10 min), and stained overnight with Coomassie colloidal blue. Excess stain was washed away with water in 3 × 10 min washes. Each band was excised from the gel and sliced into small (1–2 mm) pieces. Gel pieces were destained with 40% acetonitrile, 50 mM NH_4_HCO_3_, and then reduced with 10 mM DTE in 100 mM NH_4_HCO_3_ for 20 min at RT. Following reduction, the samples were alkylated with 54 mM iodoacetamide in 100 mM NH_4_HCO_3_ for 20 min at RT in the dark. Following that, the samples were washed with 100 mM NH_4_HCO_3_ for 20 min at RT, followed by another wash with 40% Acetonitrile, 50 mM NH_4_HCO_3_ for 20 min at RT, and a final wash with ultrapure water under the same conditions. Gel pieces were dried in a centrifugal vacuum concentrator (speed vac) and trypsinized overnight with 600 ng of trypsin per sample (trypsin stock solution: 10 ng/μL in 10 mM NH_4_HCO_3_, pH 8.5) at RT in the dark. After incubation with 50 mM NH_4_HCO_3_ for 15 min at RT, followed by two incubations with 5% formic acid:50% acetonitrile for 15 min at RT. Peptides were eluted in a final volume of 600 μL and filtered through 0.22 μm PVDF filters (Merck Millipore) before drying in a centrifugal vacuum concentrator (SpeedVac). Dried peptides were reconstituted in mobile phase A buffer (0.1% formic acid, pH 3) and analyzed by LC-MS/MS [44].

#### 4.7.2. LC-MS/MS Analysis

Samples were resuspended in a 10 μL mobile phase A buffer. A 5 μL volume was injected into a Dionex Ultimate 3000 RSLS nanoflow system (Dionex, Camberly, UK) configured with a Dionex 0.1 × 20 mm, 5 μm, 100 Å C18 nano trap column with a flow rate of 5 µL/min. The analytical column was an Acclaim PepMap C18 nano column 75 μm × 50 cm, 2 μm, 100 Å with a flow rate of 300 nL/min. The trap and analytical column were maintained at 35 °C. Mobile phase B was 100% acetonitrile:0.1% formic acid. The column was washed and re-equilibrated prior to each sample injection. The eluent was ionized using a Proxeon nanospray ESI source operating in positive ion mode. A Q Exactive Orbitrap (Thermo Finnigan, Bremen, Germany) was operated in MS/MS mode for mass spectrometry analysis. The peptides were eluted under a 240 min gradient from 2% (B) to 33% (B). Gaseous phase transition of the separated peptides was achieved with positive ion electrospray ionization applying a voltage of 2.5 kV. For every MS survey scan, the top 10 most abundant multiply charged precursor ions between *m*/*z* ratio 300 and 2200 and intensity threshold 500 counts were selected with FT mass resolution of 70,000 and subjected to HCD fragmentation. Tandem mass spectra were acquired with an FT resolution of 35,000. The normalized collision energy was set to 33, and already targeted precursors were dynamically excluded for further isolation and activation for 30 s with 5 ppm mass tolerance.

#### 4.7.3. MS Data Processing

Raw files were analyzed with the Proteome Discoverer 1.4 software package (Thermo Finnigan, ThermoFisher), using the Sequest search engine and the Uniprot human (Homo sapiens) reviewed database, downloaded on 15 December 2017, including 20,243 entries. The search was conducted using cysteine carbamidomethylation as a static modification and methionine oxidation as a dynamic modification. Two missed cleavage sites were permitted, along with a precursor mass tolerance of 10 ppm and a fragment mass tolerance of 0.05 Da. False discovery rate (FDR) validation was set to < 0.01.

### 4.8. Co-Immunoprecipitation

Forty-eight hours following transfection of approximately 1.5 × 10^7^ SK-N-SH cells with DDIT4 expressing plasmid, cells were resuspended in 2 mL of ice-cold non-denaturing PLB lysis buffer containing 10 mM HEPES pH 7.0, 100 mM KCl, 5 mM MgCl_2_, 0.5% NP-40, and protease inhibitors. After incubation for 30 min in a rotor at 4 °C, the cell suspension was centrifuged for 15 min at 14,000× *g* at 4 °C, and the supernatant was transferred into new tubes. Half of the supernatant was then incubated overnight with 2 μg of either anti-goat IgG or anti-DDIT4 antibodies at 2–8 °C with continuous mixing.

PureProteome™ Protein A/G mix magnetic bead suspension (Merck/Millipore, Burlington, MA, USA) was washed with PBS containing 0.1% Tween-20 thrice and blocked for an hour at RT with PBS containing 2% BSA and 0.1% Tween-20. Beads were then washed three times with PBS and equilibrated with NT2 washing/elution buffer containing 50 mM Tris-HCl pH 7.4, 250 mM NaCl, 1 mM MgCl_2_, and 0.05% Tween-20. Each lysate was incubated with 30 μL of bead slurry in microcentrifuge tubes with rotation for 30 min at RT. The beads were subsequently washed three times with an NT2 buffer. Immunoprecipitated proteins were then eluted by resuspending the bead slurry in 100 μL of NT2 buffer supplemented with 6× Laemmli buffer containing 250 mM Τris-HCl pH 6.8, 6% SDS, 30% β-mercaptoethanol, 40% glycerol, and 0.005% bromophenol blue for 10 min at 98 °C.

### 4.9. Gene Ontology Analysis

Gene Ontology (GO) ‘cellular component’, ‘molecular function’, and ‘biological process’ analyses were performed using the WebGestalt 2019 gene set analysis toolkit with an FDR < 0.05 [45]. The Homo sapiens genome protein-coding database was used as a reference.

To visualize the non-redundant biological terms in a functionally grouped network, the Cytoscape plugin ClueGo 2.5.8 was used [46]. The ClueGO network is created with kappa statistics and reflects the relationships between the terms based on the similarity of their associated genes. The node color is switched between functional groups and cluster distribution on the network. Related terms that share similar associated genes were fused to reduce redundancy. 

## Figures and Tables

**Figure 1 ijms-23-05189-f001:**
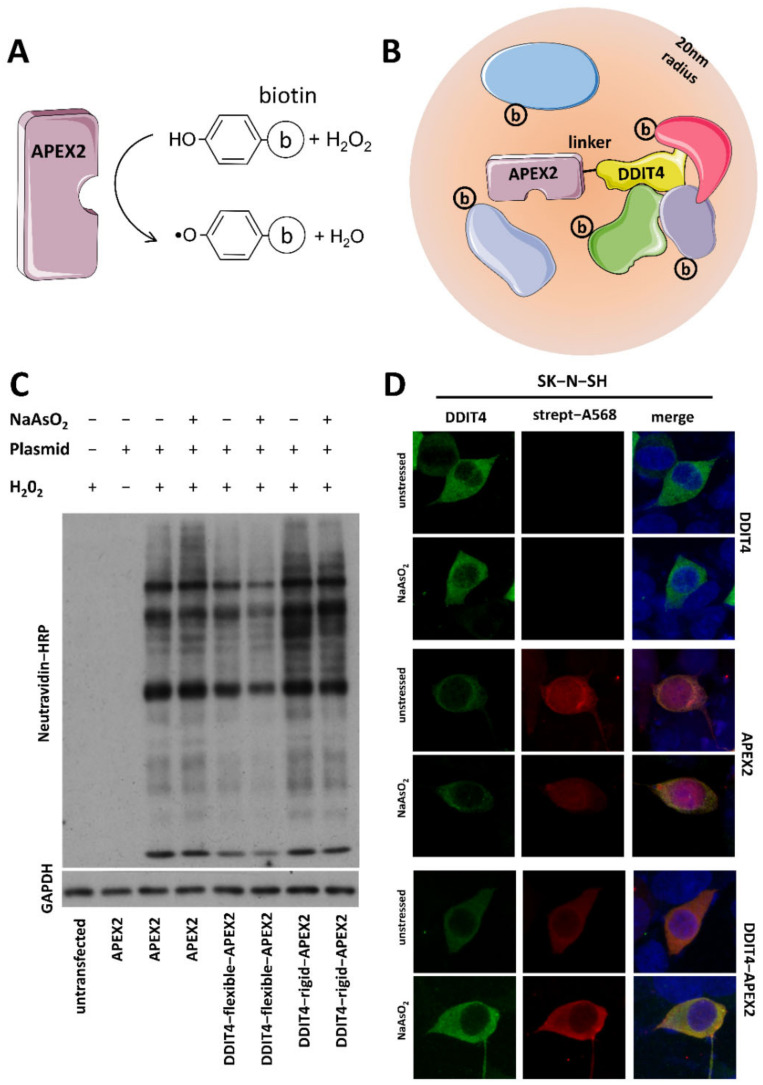
APEX2-mediated biotinylation maps DDIT4-interacting proteins (**A**) APEX2 enzyme uses H_2_O_2_ to catalyze biotin-phenol oxidation that generates short-lived biotin phenoxyl radicals that covalently tag proximal proteins; (**B**) Schematic diagram of the DDIT4–APEX2 proximity labeling depicting APEX2 tagged via a linker sequence to the C-terminus of human DDIT4. Labeled proteins are located within 20 nm of the DDIT4–APEX2 hybrid; some will interact directly with DDIT4; (**C**) NeutrAvidin-HRP Western blotting of induced protein biotinylation in lysates from cells expressing APEX2 or DDIT4–APEX2 (using different linker sequences); (**D**) Immunostaining of unstressed and sodium-arsenite-treated neuroblastoma SK-N-SH cells transfected with either DDIT4, APEX2, or DDIT4–APEX2 plasmids; (**E**) Venn diagram showing overlapping hits from four independent experiments in unstressed and acutely stressed conditions. Some elements in this image were obtained from Servier Medical Art (http://smart.servier.com/), permissible to use under a Creative Commons Attribution 3.0 Unported License.

**Figure 2 ijms-23-05189-f002:**
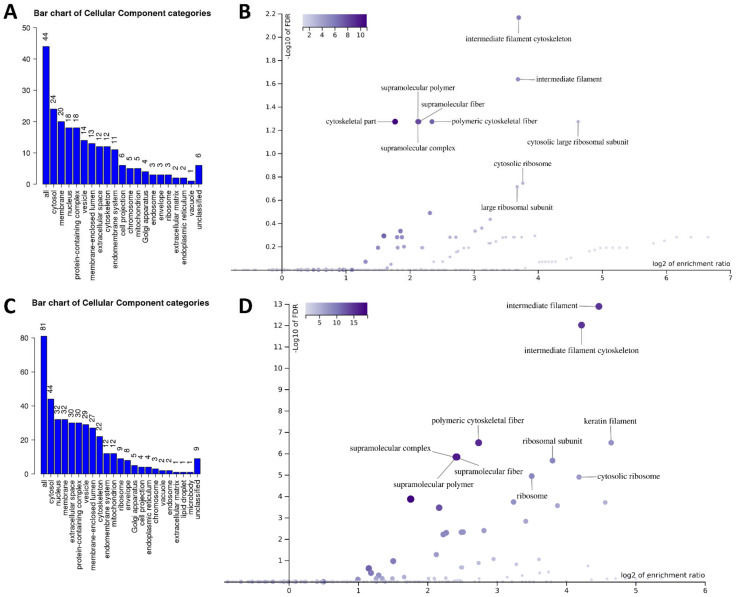
Subcellular localization of DDIT4-interacting proteins. ‘Cellular component’ categories of all DDIT4-interacting partners in unstressed (**A**) and sodium-arsenite-treated (**C**) SK-N-SH cells. The height of the bar represents the number of protein IDs in the category. Next, the ‘cellular component’ categories specifically enriched in unstressed (**B**) and sodium-arsenite-treated (**D**) SK-N-SH cells. WebGestalt analysis software was used to visualize both bar charts and volcano plots.

**Figure 3 ijms-23-05189-f003:**
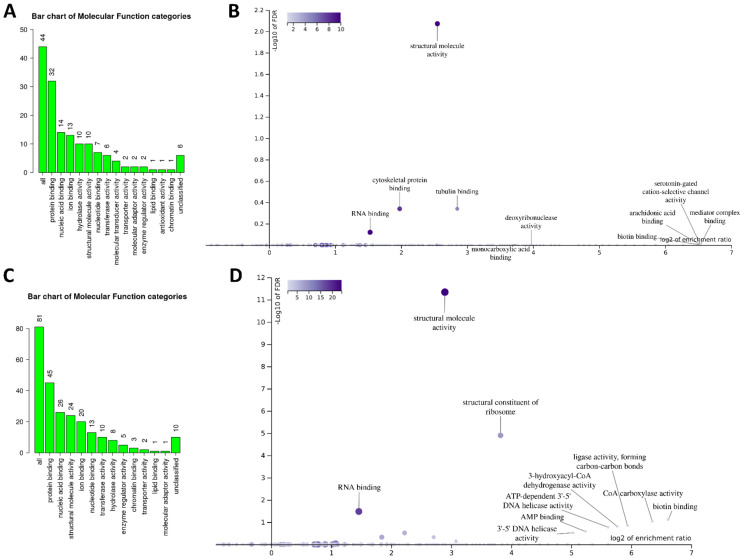
Molecular function of DDIT4-interacting proteins. ‘Molecular function’ categories of all DDIT4-interacting partners in unstressed (**A**) and sodium-arsenite-treated (**C**) SK-N-SH cells. The height of the bar represents the number of protein IDs in the category. Next, ‘molecular function’ categories specifically enriched in unstressed (**B**) and sodium-arsenite-treated (**D**) SK-N-SH cells. WebGestalt analysis software was used to visualize both bar charts and volcano plots.

**Figure 4 ijms-23-05189-f004:**
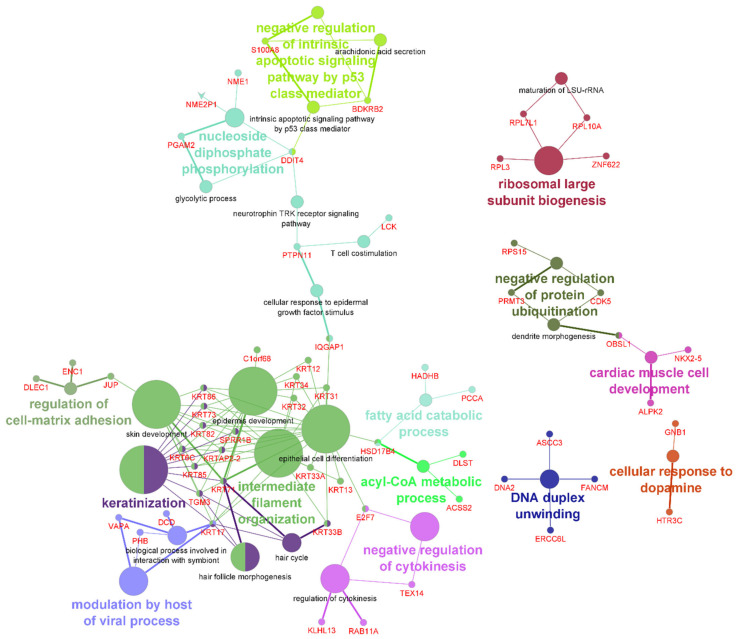
Interaction network of ‘biological processes’ obtained using the 118 unique proteins labeled by DDIT4–APEX2 in unstressed and sodium-arsenite-treated cells. GO terms are represented as nodes, and the node size represents the term enrichment significance. The color of the nodes changes depending on the functional groupings. The network was visualized in Cytoscape running the ClueGo plugin using GO term fusion, kappa score 0.4, and a yFiles organic layout.

**Figure 5 ijms-23-05189-f005:**
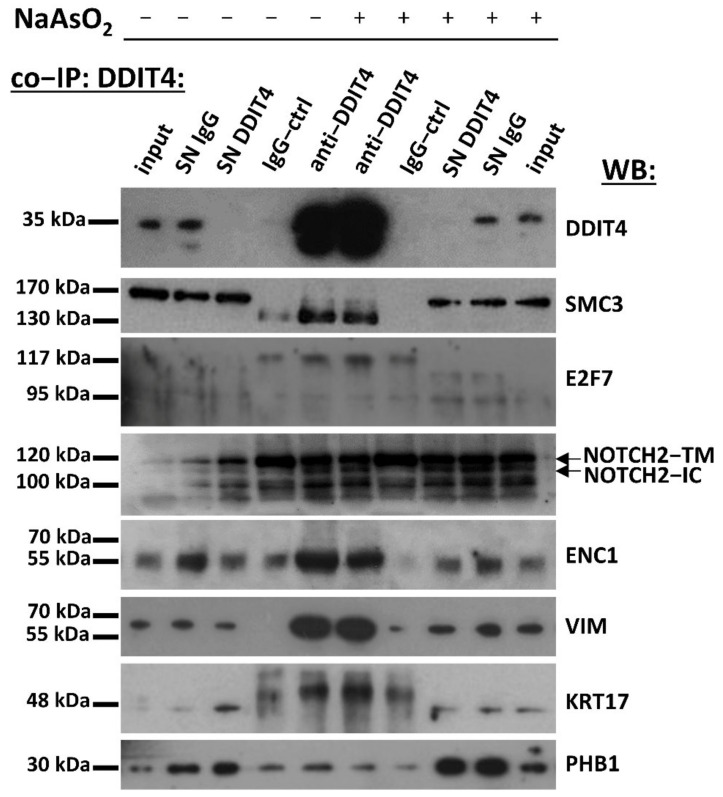
Several protein partners identified by proximity biotinylation were found to interact with DDIT4 directly. DDIT4-overexpressing SK-N-SH cells treated with or without sodium arsenite were subjected to co-immunoprecipitation reactions using anti-DDIT4 antibody or IgG as a control. Precipitated proteins and the original lysates (inputs) were analyzed by SDS-PAGE followed by immunoblotting detecting DDIT4, SMC3, E2F7, NOTCH2, ENC1, VIM, KRT17, PHB1. SN: supernatant of the lysate after immunoprecipitation, TM: transmembrane, IC: intracellular.

**Table 1 ijms-23-05189-t001:** ‘Biological process’ annotation of proteins labeled in DDIT4–APEX2 reactions. The table presents the results of the *ClueGO* ‘biological process’ analysis. Nr.: Number of genes associated with the GO term. %: Genes identified as a percentage of all linked genes in GO. PVal: *p*-value of the GO term after Benjamini–Hochberg correction.

GO BP Term	Term PValue	% Associated Genes	Nr. Genes	Genes Cluster #1: Control Condition	Genes Cluster #2: Acute Stress Condition
Intermediate filament organization	1.5 × 10^−6^	18.31	13	[KRT13, KRT17, KRT71]	[KRT13, KRT31, KRT32, KRT33A, KRT33B, KRT34, KRT6C, KRT73, KRT82, KRT85, KRT86]
Cytoplasmic translation	2.7 × 10^−6^	5.75	10	[EIF3G, RPL27, RPL32]	[FAU, RPL10A, RPL13A, RPL3, RPL37A, RPS15, RPS17]
Glycolytic process	1.2 × 10^−2^	2.08	2	[DDIT4]	[DDIT4, PGAM2]
Ribosomal large subunit biogenesis	2.3 × 10^−2^	5.33	4	[ZNF622]	[RPL10A, RPL3, RPL7L1]
Negative regulation of cytokinesis	8.0 × 10^−2^	28.57	2	[-]	[E2F7, TEX14]
Hindbrain morphogenesis	8.2 × 10^−2^	5.45	3	[CDK5]	[DLEC1, PTPN11]
Modulation by host of viral process	8.6 × 10^−2^	7.69	3	[KRT17]	[PHB, VAPA]
DNA duplex unwinding	8.6 × 10^−2^	3.96	4	[DNA2]	[ASCC3, ERCC6L, FANCM]
Response to dopamine	1.2 × 10^−1^	2.08	2	[GNB1, HTR3C]	[-]
Negative regulation of innate immune response	1.6 × 10^−1^	2.67	2	[ADAR, LYAR]	[-]
Regulation of cardiac muscle contraction	1.6 × 10^−1^	2.47	2	[-]	[JUP, NKX2-5]
Branched-chain amino acid metabolic process	1.7 × 10^−1^	6.45	2	[PCCA]	[MCCC1, PCCA]
Activation of cysteine-type endopeptidase activity involved in apoptotic process	1.7 × 10^−1^	3.41	3	[S100A8]	[DLEC1, LCK]
Cardiac muscle cell development	1.8 × 10^−1^	3.80	3	[OBSL1]	[ALPK2, NKX2-5]
Neuron projection organization	1.9 × 10^−1^	2.08	2	[CDK5, PRMT3]	[-]
Fatty acid beta-oxidation	2.1 × 10^−1^	2.41	2	[-]	[HADHB, HSD17B4]
peptide cross-linking	2.1 × 10^−1^	5.26	2	[SPRR1B, TGM3]	[TGM3]
Deoxyribonuclease activity	2.2 × 10^−1^	2.82	2	[DNA2, NME1]	[-]
ATP generation from ADP	2.3 × 10^−1^	2.06	2	[DDIT4]	[DDIT4, PGAM2]
Acetyl-CoA metabolic process	2.4 × 10^−1^	5.41	2	[-]	[ACSS2, DLST]
Positive regulation of mitotic cell cycle phase transition	2.6 × 10^−1^	2.15	2	[ESPL1, RAB11A]	[-]
Ribosomal small subunit biogenesis	2.8 × 10^−1^	2.67	2	[-]	[RPS15, RPS17]

## Data Availability

Data is contained within the article or Appendix A.

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
