# Peer review of "APEX2-Mediated Proximity Labeling Resolves the DDIT4-Interacting Proteome"

_ijms, 2022, doi:10.3390/ijms23095189_

Round 1

Reviewer 1 Report

The authors describes an interesting approach based on APEX labeling method to evaluate the proteome proximity interactors of DDIT4 in unstressed and acute stress conditions, in order to gain information to explore molecular mechanisms underlying the function of this stress response transcript.

Few points must be clarified before their publication especially regarding the methodology:

1_It is not clear to the reviewer how many runs/technical replicates have been performed for each sample for LC-MS/MS analysis. At page 13: it seems that each sample has been analyzed once! Please add the number of replicates or justify this issue.

2_ The difference in number of non-specific APEX2 proteins in the two studied conditions is considerable (571 vs 446) (+28%).  “ In total, 316 proteins were detected across DDIT4-APEX2 experiments by MS in the unstressed condition, and after filtering out the dataset for non-specific labeling by APEX2 alone (571 proteins), 44 proteins were unique to DDIT4 (Supplementary Table S2). In the acute stressed condition, DDIT4-APEX2 labeled a similar number of proteins (337 proteins), and after filtering out the APEX2 dataset (446 proteins), 81 proteins were unique to DDIT4 (Table S1).” Please comment this point to exclude a possible bias due to the technical variability.

3_Related to Figure 1C. The authors should clarify why the proteins obtained with the rigid linker DDIT4_rigid_APEX2 (++ and +++) are more abundant (and probably more) than APEX2 (++ and +++) and DDIT4_flexible_APEX2 (++ and +++). Have you a possible explanation?  Have you analyzed these samples by LC-MS to get more quantitative and qualitative data about their difference?

4_Page 9: “In terms of the data itself, there was a quantitative but not functional distinction between the proteomes obtained in unstressed and acute stressed conditions. During acute stress, DDIT4 had twice as many interacting partners. Further, although there was minimal overlap between the two protein lists,”. The authors should provide data or better justification about these differences (why the identified proteins in the two conditions are different?). Could this difference be related to the possible variability described in point 3?

Reviewer 2 Report

In the present study, the authors developed and optimized an efficient labeling system for the protein of interest (DDIT4) interactome and applied this approach to the stress model of sodium arsenite in SK-N-SH glioblastoma cells. The method seems to be convenient and efficient (moreover, it has already been successfully tested by this team in a previous work - Gourdomichali et al., 2022).

Nevertheless, there are some questions for the authors, in particular, methodological ones. The in-gel-trypsinolysis leads to the inevitable loss of material, which can be critical in the case of immunoprecipitation. I suppose the removal of PEG-containing detergent from the immunoprecipitates (especially since this is not a problem at all using magnetic beads), followed by trypsinolysis in solution to be more efficient. Moreover, the capabilities of the equipment used by the authors make it possible to avoid additional fractionation of the protein samples by individual bands excision.

p.14, lines 1-2: "6x Laemli buffer containing 6% SDS, 30% β-mercaptoethanol, 40% glycerol, and 0.005% bromophenol blue" - Laemmli (not Laemli) buffer usually contains Tris-HCl, pH 6.8
